# What works, how and in which contexts when using digital health to support parents/carers to implement intensive speech and language therapy at home for children with speech sound disorder? A realist review

Naomi Leafe[1]*, Emma Pagnamenta[2], Laurence Taggart[1], Mark Donnelly[3], Angela Hassiotis[4], Jill Titterington[1]

1 Institute of Nursing and Health Research, Ulster University, Belfast, United Kingdom, 2 School of Psychology and Clinical Language Sciences, University of Reading, Reading, United Kingdom, 3 School of Computing, Ulster University, Belfast, United Kingdom, 4 Division of Psychiatry, University College London, London, United Kingdom

* leafe-n@ulster.ac.uk

## Abstract

### Purpose

Digital health solutions to support parent-implemented interventions alongside direct speech and language therapist (SLT) input could help increase intervention intensity for children with speech sound disorder (SSD) to meet evidence-based recommendations. This realist review explores the factors which could make intensive parent-implemented digital interventions for children with SSD effective, and how this complex intervention might work in different contexts.

### Methods

Realist review methodology was adopted to explore what works, why, how, for which parents/carers, and in what circumstances. Realist methods aimed to understand the active ingredients, contexts, and associated outcomes of this complex intervention. Preliminary theories were developed to describe how and why digital parent-implemented interventions work for children with SSD. Data was extracted from 43 papers to test and refine preliminary theories. Behaviour change theories were used to explain how the intervention works in practice.

### Results

A set of 20 explanatory theories were developed to depict how and why digital parent-implemented interventions work in different contexts. Theories covered five areas: child-participation; the child-parent-SLT dynamic; parent-training; partnership and collaboration; intervention intensity. The theories describe mechanisms of the

**Data availability statement:** The data supporting the findings reported in this paper are openly available from Ulster University's Research Portal at https://doi.org/10.21251/2b0e8c5c-f334-4090-a682-56cd71c75ab4

**Funding:** All the funding for this review has come from a PhD studentship at Ulster University (NL) supported by the Department for the Economy (DfE). The funders had no role in study design, data collection and analysis, decision to publish, or preparation of the manuscript. There was no additional external funding received for this study.

**Competing interests:** The authors have declared that no competing interests exist.

intervention and how these are responded to in different situations. Findings highlight the importance of intensive intervention for children with SSD.

## Conclusions

This realist review adds new in-depth insight into how digital parent-implemented interventions work, for whom, and why. This new understanding has potential to support future successful digital parent-implemented interventions and increase intervention intensity for children with SSD globally. Implications for services and the potential of emerging digital health approaches to promote parent-implemented interventions are discussed.

## Introduction

### Intervention intensity for speech sound disorder (SSD)

SSD is defined as any combination of difficulties with perception, articulation/motor production, and/or phonological representation of consonants and vowels, syllable and word shapes, and prosody that may impact speech intelligibility and acceptability [1]). Community cohort studies have reported a prevalence of SSD of 3–4% in 4–6 year old children [2,3]. Whilst the nature and severity of SSD varies, children with moderate to severe SSD who cannot be easily understood require specialist speech and language therapy intervention [4] and are at risk of poorer literacy, educational attainment and social and emotional well-being outcomes [5,6]. The use of terminology relating to severity in this paper is based on the criterion for standard scores. A standard score of <70 (percentile rank (PR) <2) indicates severe SSD, a standard score of 71–77 (PR 3–6) indicates moderate SSD, and standard scores 78–85 (PR 7–16) would be classed as mild SSD. In clinical practice, these scores would be considered in the context of the functional impact of the child's SSD on activity and participation [4,7].

The importance of intervention intensity for children with SSD is becoming more widely recognised globally [4–6,8–12]. Warren, Fey, and Yoder [13] proposed terms to define the components of intervention intensity in communication disorders: dose (e.g., number of teaching episodes per session); dose form (e.g., treatment approach); dose frequency (how often sessions are provided); and total intervention duration (the time over which intervention is provided). The product of these components provides an abstract measure of the total treatment intensity (the cumulative intervention intensity (CII)) [13]. While the optimum CII for phonological interventions is unclear as yet, research suggests that higher intensities are more effective and efficient [9]. When considering children with moderate to severe SSD, the evidence-base recommends 2–4 sessions per week, lasting 30–60 minutes, with at least 70−100+ trials per session [4–6,8–10,12,14]. At this stage findings vary about the total intervention duration required for children with SSD, ranging from the requirement for more than 8 weeks of therapy to effect change [15] to 30–40

sessions for children with phonological disorder [16] and up to 104+ hours depending on the nature and severity of the child's SSD [10,17]. The severity of the child's SSD underpins the potential need for a higher overall CII.

## Evidence-practice gap

Gaps between evidence-based recommendations for intensity of intervention and practice have been found in speech and language therapy services worldwide (see Table 1) [12,18–20,22].

While the length of sessions reported in these studies generally complies with the evidence-base, the frequency of sessions generally does not unless the context supports this, e.g., certain preschool settings in the United States [18]. Dose has generally not been investigated, and where it has, it either falls vastly below the evidence-based recommendations of at least 50 trials for children with mild-moderate difficulties and 70−100+ for children with more severe difficulties [10,16], or just tips into the more evidence-based range [12,19]. The number of sessions provided for the total intervention duration is notably variable across countries.

Many SLT services also incorporate breaks from ongoing intervention into their models of delivery (often to manage clinical resources) [12] and it is unclear how this stop-start model of intervention delivery impacts interpretation (or indeed impact) of the CII received. For example in the UK, community-based intervention is often delivered in 6-week blocks [11]. As was found by McFaul et al. [11] these breaks can have a negative impact on learning as a child waits, sometimes for months, before looping back into the service [19].

Clearly, current models of service provision challenge the delivery of recommended intensities of intervention worldwide. Whilst SLTs have some flexibility around dose per session, caseload sizes and funding impact SLT control over how frequently children are seen, for how many sessions, and how long [9,11,12,19,23]. Providing 2–4 sessions per week can be particularly challenging in community SLT services, whilst managers, specialists and SLTs employed in private practice may have more flexibility [19,24]. These differences in services and working contexts create considerable discrepancies in services offered to families.

**Table 1. Summary of findings from key surveys investigating intensity of SLT intervention for children with SSD in current practice across countries.**

| Authors | Country | First language of country | Age of children | Dose (# of target trials) | Frequency per week | Session length | Total intervention duration |
|---|---|---|---|---|---|---|---|
| Brumbaugh and Smit [18] | United States | English | 3–6 yrs | – | 1–2 weekly | 15–30 minutes | – |
| Hegarty et al. [19] | United Kingdom | English | 4 yrs 8 mths | Mostly 10–30 per session | Mostly x1 weekly | Mostly 21–30 minutes | Mostly 9–12 sessions |
| Oliviera et al. [20] | Portugal | Portuguese | – | – | Mostly x1 weekly | – | Mostly 6+ months |
| Sugden et al. [12] | Australia | English | Mean of 4 yrs 6 mths | 21–49 (37.6%)/50–99 (39.8%) per session | Mostly x1 weekly | Mostly 30–44 minutes | 2–400 sessions (mean 22.7) |
| To et al. [21] | Hong Kong | Cantonese | 2–16 yrs | – | Public settings: Mostly x1 a fortnight Private settings: Up to x1 weekly | Public settings: Mostly 30–35 minutes Private settings: Mostly 30–60 minutes | Public settings: Mostly 1–20 sessions Private settings: Mostly 5–12 sessions |

*Note.* Dashes (as relevant to the column header) either indicate that the component of intensity was not investigated or that the age of children worked with by the surveyed SLTs was not specified. Information on SES and the first language of children receiving intervention from SLTs, was not covered in these surveys, apart from Sugden et al. [12] who reported that most SLTs they surveyed had caseloads containing < 10% of children from culturally and linguistically diverse backgrounds.

Family factors can also impact access to intensive intervention. Attending twice to four-times weekly appointments requires high commitment from parents/carers, presenting challenges such as travel (potentially being too expensive for some families), taking time from work and/or school, sickness, and child fatigue [11]. These factors potentially impact on the accessibility and equity of intensive services. Alighieri et al. [25] explored the acceptability of high or low intensity SLT intervention with 12 parents of children with cleft (lip and) palate. Parents recognised benefits of both intensities of intervention offered, but labelled factors associated with accessing more frequent appointments as burdensome, such as travel, taking time off work, and arranging sessions to fit family schedules. Despite this, parents concluded that high-frequency therapy is less burdensome overall as the total intervention duration is likely to be shorter.

If higher intensity of intervention is more efficient and effective for children with SSD, then alternative, inclusive ways of supporting increased intensity in current service contexts need to be explored. Intensive intervention needs to be acceptable and accessible to children and families [26]. Involving parents/carers in implementing intervention at home has been recognised as one possible solution to help increase the intensity and effectiveness of therapy [9,25,27–31].

## Parent-implemented interventions

Parent/carer involvement in intervention under guidance of an SLT, referred to here as parent-implemented interventions, has been found to support positive outcomes for children with speech, language, or communication needs (SLCN) [31–34]. Parents/carers supporting home-intervention facilitates more regular practise, potentially increasing the intensity of intervention children receive [27,28,31].

Parents/carers can be successful implementers of SSD intervention alongside direct SLT sessions when supported, trained, and empowered by SLTs [14,30,35]. A randomised controlled trial [30] involving 44 children with cleft palate, compared the use of Parent Led, Therapist Supervised, Articulation Therapy (PLAT) with typical service delivery from a SLT. They found that PLAT can be as effective as routine SLT-implemented intervention in facilitating positive change to children's outcomes. Sugden et al. [14] evaluated parent/carer and SLT combined intervention for five children with phonologically based SSD. Results showed that parents/carers can deliver therapy at home with competence when supported, and that parental involvement could fulfil a gap in the intensity of intervention provided for children with SSD. However, they acknowledged that the specific characteristics of parents/carers and children that optimise success of parent-implemented intervention needs further study.

Contextual factors are recognised to impact on the effectiveness of parent-implemented interventions for SSD; what works for one family may not work for another [29,30,36]. Studies have reflected on factors potentially influencing outcomes, including parental confidence and skills, treatment fidelity, and child factors [25,29,37]. Further exploration is needed for deeper understanding of the specific contexts in which parent-implemented interventions may work, for whom, and why.

## Digital health

Digital delivery of services can be used to train, upskill, and support parents/carers to facilitate parent-implemented intervention [25,36]. The term digital health encompasses different types of information and communication technology in healthcare, which may support diagnosis, monitoring, treatment, or self-care, as well as staff training or clinical decision-making [38]. The terminology surrounding digital health varies, and in this review, the term 'digital tool' will be used to describe digital health platforms used to support speech and language therapy interventions.

## Digital tools in speech and language therapy interventions

Health services are increasingly adopting digital tools to facilitate evidence-based interventions. Digital tools may reduce contextual challenges faced by families accessing in-person appointments by reducing travel, increasing convenience of appointment scheduling, reducing time commitments, and facilitating access to social support networks [25,39,40].

In speech and language therapy, digital tools can facilitate intervention delivery, staff training, and parent/carer support. "Supporting Understanding of Speech Sound Disorder" (SuSSD) [41] is an example of a digital tool which supports evidence-based interventions for children with SSD. SuSSD, an online manualised evidence-based tool co-produced with SLTs, supports SLTs to appropriately identify and deliver evidence-based phonological interventions for children with SSD, offering a digital solution to reducing the evidence-practice gap.

Digital tools to support and train parents/carers to facilitate home-intervention have also been explored [39]. Bellon-Harn et al. [36] carried out a scoping review of peer-reviewed journal articles on the use of videos/digital media in parent-implemented interventions for children with primary SSD or language disorder. Findings suggested that these platforms can increase access to parent-implemented interventions and help parents/carers feel more competent and empowered. However overall, findings highlighted limited evidence, with no studies focusing on children with SSD. Further exploration is needed into how digital tools facilitate parental learning and empowerment, under which circumstances, when considering parent-implemented interventions for SSD.

Digital tools that incorporate digital games, referred to as Gamification [42], can also support children with SSD. Digital games are motivating to children, with the potential to increase the frequency of home-practice in parent-implemented intervention, enhancing SLT support [43–45]. Studies have identified factors that can impact on children's responses to digital games in therapy, such as age, interests, and task difficulty [27,43,44,46]. It would be beneficial to continue to build understanding of the underlying components and related contextual factors of games in digital tools impacting on their successful use in home-intervention for SSD.

### Summary

Parent-implemented interventions offer a potential solution to challenges in providing recommended intervention intensity for children with SSD, particularly dose and frequency, to improve efficiency and effectiveness of services and child outcomes. Digital tools offer a platform to enable and enhance parent-implemented interventions, including supporting parent-training and motivating children. Further exploration of the active ingredients involved and the contexts that would enhance certain outcomes (or not) of digital, parent-implemented interventions for SSD is needed to understand how this complex intervention may work in practice, for whom.

### Aim of paper

To synthesise relevant literature to understand what works, how, for whom, and in what contexts, to support delivery of an effective, intensive digital parent-implemented intervention for children with SSD.

### Objectives

1. To explore why, how and in what circumstances, parent-implemented speech interventions are effective for some children and families (with a focus on supporting children with SSD).

2. To understand what factors specific to digital tools for children with SSD, enhance the effectiveness of interventions, for which children and families, in what circumstances, why and how.

3. To develop theoretical explanations and an explanatory model, which captures how to empower parents/carers to deliver effective, intensive speech intervention supported by a digital tool, at home for their child with SSD.

### Method

This realist review followed methods based on guidelines by Pawson [47,48] and Hunter et al. [49] and was carried out and reported in line with guidance and publication standards by RAMESES (Realist And MEta-narrative Evidence

Syntheses: Evolving Standards) [50,51]. Realist review methodology was chosen as a theory-driven approach that would help develop insight into the complexity of digital, intensive, parent-implemented interventions for children with SSD. This realist review followed six stages, as shown in S1 Fig. Definitions of terminology used in realist methods can be found in Table 2. The method for the realist review is summarised in Table 3, and can be found in detail in the published protocol [56].

Realist studies strive to uncover what is happening underneath the surface to produce observable outcomes; known as generative causation [49]. In realist methodology, we need to understand the underlying *mechanisms* that lead to outcomes in an intervention, that is to understand the resources an intervention provides and how people respond to them. We also need to understand the *context* of the intervention and how this context interacts with mechanisms to lead to certain *outcomes* [49].

To help depict this theoretical thinking about the link between contexts (C), mechanisms (M), and outcomes (O) of interventions, realist studies involve the development of CMO configurations based on thinking harvested from the data within published papers. For example, in a certain situation (C), implementing an aspect of an intervention (M), will be responded to by those receiving the intervention (M), to create a certain result (O). An example may be: a child with SSD needing a high dose of intervention (C). Being given the opportunity to earn points in a game for producing a target trial (M - resource), motivates them to repeat target trials (M - response), leading to progress with their target (O). In realist science the mechanism is made up of two uniquely combined components - the resource which is what is provided to change the behaviour and the response to that which results in the outcome dependent on context.

The way different contexts interact with mechanisms is not linear. This interaction is described and supported by individual 'funnel' CMO models in the results of the review (see example model in S2 Fig). In the models, the funnel shows the intervention mechanisms, with the yellow boxes showing resources provided by the intervention and the pink circles showing potential responses from those receiving the intervention. The potential outcomes of these mechanisms are captured in the green rectangle coming out of the funnel. Contexts are layered around the funnel to show how these impact on the way the mechanism/s work, leading to different outcomes.

**Table 2. Definitions of realist terms.**

| Term | Definition |
|------|-----------|
| Programme architecture | The formal and informal components that make up an intervention (e.g., parent/carer training in parent-implemented interventions). Formal components might be those which are advised, and informal may be how components are adapted in clinical practice [52] |
| Context | The circumstances surrounding the intervention that influence the way an intervention is delivered and responded to. Contexts can be observed across different levels and types (e.g., personal, geographical, social, physical) |
| Mechanism | An unobservable force created by an intervention that leads to outcomes. Mechanisms are the intended or unintended resources that are offered in an intervention (e.g., empathy), which leads to a response from those involved (e.g., increased self-worth). Mechanisms help us to understand why and how an intervention works [49,53] |
| Outcome | The intended or unintended result of an intervention (e.g., parent/carer engagement) |
| Programme theory | Ideas that explain what makes a programme/ intervention work, how, and why, in certain contexts |
| Initial rough programme theory (IRPT) | Early hypothetical ideas at the beginning of a realist study that outline contexts, mechanisms, and outcomes to explain how and why the programme works [52] |
| Context-mechanism-outcome configurations (CMOs) | CMOs are a way of depicting how contexts, mechanisms, and outcomes of an intervention react. CMOs show causal explanations about outcomes of an intervention, to explain what works, for who, how, in what contexts. CMOs can be about parts of an intervention or the whole intervention [53] |
| Middle-range theory | A level of explanatory theory that can be used to explain the logic of how parts of an intervention work when implemented. MRT is abstract to the point it can be applied across different interventions in a range of settings, but tangible enough that it can be tested in relation to the implementation of the intervention being explored [54]. Existing MRT may help develop theoretical thinking, for example Normalization Process Theory [55] may help add depth to how processes are implemented and embedded in practice. |

**Table 3. Summary of the stages of the realist review and methods used.**

| Stage | Methods |
|---|---|
| 1.Identifying and developing the research question | Scope of topic and key components of the intervention identified through informal review of literature and expertise of research team (all) |
| | PICO framework used to identify key aspects of the intervention to be explored (NL) |
| 2.Developing Initial Rough Programme Theories | Developed initial ideas about how the intervention might work using exploration of the literature and expert experience. |
| | Initial Rough Programme Theories (IRPTs) formed to explain how the intervention works through initial scoping of the literature, expert experience, and team discussions. IRPTs were organised around four aspects: the digital tool, child, parent/carer role, and SLT role. The first iteration of IRPTs were shared with an expert steering group, leading to new theories, and rewording or refinement of existing theories (NL, JT, EP) |
| | A set of IRPTs were finalised to be taken forward for review. An example of an IRPT included: by being able customise intervention activities to the child on a website or app (mechanism resource), the content will feel personal, meaningful and relatable to the child (mechanism response), which will help motivate them to engage with the activities (outcome) (IRPTs can be accessed by contacting the author).<br>A preliminary model to show early explanatory thinking was developed |
| 3.Developing a formal search strategy | Search terms were created to explore the IRPTs (all), reviewed by the expert steering group (see S3 Table) |
| | Four databases were searched in May 2022: Scopus, CINAHL, Web of Science, Medline (NL, supported by a subject librarian). An updated search was completed in April 2024. |
| 4.Selection and quality appraisal of evidence | Three bespoke tools were used to screen papers for relevance and rigour (as in protocol, Leafe et al. [56]): |
| | Article abstract screening using identification tool, to identify the relevance to speech and language therapy intervention for children aged 2–7 years with SSD, and relevance to parental involvement, digital intervention, and/or intervention intensity. Studies were not included beyond this stage if they fell within the exclusion criteria, i.e., the population receiving intervention had an additional developmental or sensory diagnosis (the full exclusion criteria is found in the published protocol paper [56]). |
| | Full text screening completed using selection tool, to screen for insight into theory development |
| | Full text screening using a bespoke appraisal tool and one of three published quality appraisal tools of methodological rigour as appropriate to the study design (Mixed Methods Appraisal Tool [57], the Single Case Reporting guideline In BEhavioural interventions checklist [58], and the PRISMA checklist [59]), to screen for rigour and robustness (NL) |
| | A random selection of papers was screened by JT and EP at each stage: Identification, 10% screened with 94% agreement; selection, 10% screened with 91% agreement; appraisal, 26% screened with 60% agreement. Appraisal and data extraction occurred simultaneously, and the appraisal stage is partly around mining for theory with higher discrepancy expected. Discrepancies between authors were not significant (e.g., one author unsure), and were discussed and resolved through discussion. |
| | Discrepancies or articles that required a second opinion were managed through second and third author screening, and discussion between authors (NL, JT, EP) |
| | A clear record of each stage was documented in Microsoft Excel ©. |
| | Additional relevant papers were identified as the study progressed through citation tracking. |
| | Selection and appraisal of the articles retrieved through the updated search was completed by NL, JT, EP, and MD |
| 5.Extracting the data | Articles were imported into and managed through NVivo © |
| | Papers were read and re-read. Data was coded based on IRPTs, and domains and intervention functions of the Theoretical Domains Framework (TDF) [60,61] |
| | A data extraction tool was used to extract key study information and relevant insight, documented in Microsoft Excel ©<br>Quotes and data extracts collected to support key insights |
| 6.Data analysis and synthesis | Extracted data, theoretically informed inferences, and middle-range theory (overarching explanations), were used to refine, refute, or confirm IRPTs (NL) |
| | New CMO configurations were created and discussed at different time points with the specialist research team (all)<br>Developed CMOs and relevant data extracts were discussed as a team for refinement and validation, to confirm, refine, and further develop theories and inferences, completed through several cycles at different timepoints (JT, EP, MD) |
| | NL and JT reviewed all CMOs and data extracts together at two timepoints to develop and confirm programme areas. CMO content, inferences, and wording were reviewed and refined through detailed discussion of data extracts and middle range theory (NL, JT). The explanatory model showing the theories underpinning the intervention was updated and refined. |

CMOs are evidence-informed, where evidence is combined with subject expertise and inferences to explain how the intervention works in different contexts [62]. This involves abductive and retroductive thinking, i.e., making inferences about how it might work, and identifying and testing 'hidden' parts of the intervention from the literature [49,52]. Middle Range Theory (MRT) (overarching theory, which generally applies to more than one CMO) is used in realist reviews (RRs) to underpin and expand explanatory theoretical thinking about how and why mechanisms lead to certain outcomes in specific contexts [48,54]. From the start of the review, including during the selection, appraisal, and analysis stages (see Leafe et al., [56]), explicit or implicit references to relevant MRTs were noted, then explored in depth, and tested using evidence extracted from the literature. This helped build understanding of the underlying mechanisms of the intervention in implementation.

In this review, initial rough programme theories (IRPTs) were tested using evidence from the literature, and either refined, accepted, or refuted. Theory development was checked and discussed with an expert steering group, and key theories were taken forward for further review. CMOs were developed iteratively throughout data identification, selection, appraisal, and extraction, supported by four bespoke tools (see Table 3, point 4 for further detail, and the study protocol for bespoke tools [56])). This four-step process ensured that data was only included if it was relevant, rich, and robust enough to support theory development. Through this iterative process, IRPTs were refined or confirmed, CMOs were amalgamated, or new CMOs were developed to create a final set of underpinning theories to explain how the intervention works in different contexts. These theories were used to develop a final theoretical explanatory model about the intervention. The protocol paper (Leafe et al. [56]) outlines detailed methods on theory development, summarised in Table 3.

### Patient and public involvement and engagement (PPIE)

An expert steering group involved in the wider study supported meaningful and useful findings from the realist review. Members of the steering group were either those involved in delivering intervention, or with lived experience or expertise relevant to the topic. This included representation from speech and language therapists, parents/carers of children with SSD, and clinical psychology. The expert steering group were involved in reviewing and refining the IRPTs to be taken forward in the realist review. Based on their input, preliminary IRPTs were refined, including rewording or amending existing IRPTs or adding new theories to be taken forward in the review, as described in Table 3. Steering group members reviewed the search terms to explore the IRPTs. The group will also be involved in dissemination of findings from the review.

## Results

### Literature characteristics

Fig 1 shows the flow diagram for article selection and inclusion, adapted from RAMESES standards and similar realist reviews [50,63–65] (see also Leafe et al's protocol paper [56]). A total of 726 articles were identified through database searches, 419 abstracts were screened, 126 articles read in full, 47 articles quality appraised, and 35 articles were included for data extraction and analysis. A further 29 citations were identified from citation tracking which were screened for identification, 28 screened for selection, 14 quality-appraised, with 8 included in data extraction. In total, 43 articles were used to inform programme theory development [4,9,11,14,23–31,36,37,43–45,66–90]. (See S4 Table for a table of included studies and a description of key characteristics). Most articles added insight into parent-implemented interventions (n=12), or digital health interventions (n=9), with 7 studies looking at intensity of intervention for SSD. Fifteen studies focused on a combination of more than one of these areas.

### Main findings

This section will cover: 1) Core middle-range theories used to develop explanatory thinking; 2) Context-mechanisms-outcome configurations; and 3) The explanatory model.

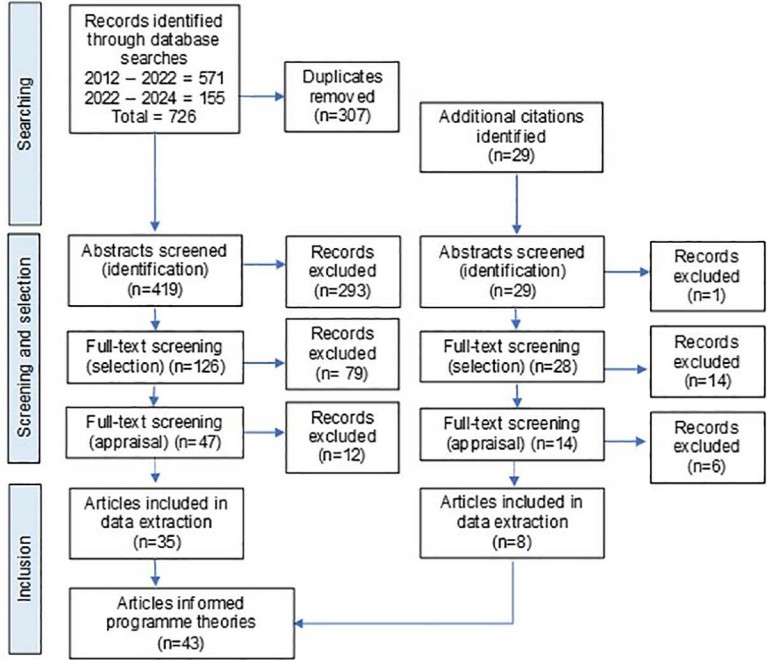

**Fig 1. Flow diagram for article searches and selection process.**

**Core MRTs used to develop and frame explanatory thinking.** The MRT theories identified and developed in this review related to behaviour change (using the Behaviour Change Wheel (BCW) as an overarching framework [91]), self-efficacy theory [92], and adult learning theory [92]. The Theoretical Domains Framework (TDF) [60,61] is a culmination of theories about factors that influence behaviour in interventions, organised across key constructs and grouped into domains. In this RR, the TDF provided a theoretical lens to support explanatory and inferential thinking and acted as a framework for identifying key mechanisms and contextual factors linked to behaviour change during data extraction and synthesis. The TDF identifies specific intervention functions which influence behaviour change. Data extracted from the included papers was mapped to these intervention functions, helping to develop, refine, and synthesise explanatory theories emerging from the primary data.

Self-efficacy theory [92] was an identified MRT with core relevance to the intervention and was explored independently to underpin theory development. Self-efficacy is defined as a person's views of their capability to achieve a goal. In the theory of self-efficacy, a person's view of their capability influences their behaviours, motivations, efforts, and resilience. A person with greater belief in their ability is more likely to succeed towards a goal. Adult-learning theory [93] was identified in this RR to explain mechanisms and outcomes of parent-implemented interventions. This theory is based on a set of assumptions thought to influence how adults learn, including their concept of themselves as a learner, readiness to learn, previous experiences, motivation, and knowing why and how learning solves a problem. These MRTs underpin key CMOs outlined in the results below and will be referenced and considered in depth in the discussion of this paper.

**CMOs (context-mechanism-outcome configurations).** The realist synthesis of the literature resulted in a set of 20 evidence-informed CMOs which explain how the intervention may work in certain contexts. These CMOs have been organised into five overarching programme areas, including:

1. Child-participation

2. The child-parent-SLT dynamic

3. Parent-training

4. Partnership and collaboration

5. Intervention intensity

The digital tool, child, parent/carer, and SLT roles are considered across each theory area. Importantly, our review found that whilst digital tools act as a platform and facilitator of interventions, the theoretical underpinning of the intervention itself remains paramount. Contextual factors and key mechanisms specific to digital tools have been considered throughout, but our theoretical thinking shows how key mechanisms impact on outcomes in parent-implemented interventions irrespective of the platform used. This will be described in the results.

**The explanatory model.** The final explanatory model (see Fig 2) shows the overarching theories that underpin successful digital parent-implemented interventions for children with SSD. The five key programme areas are shown in the cogs, which move and interact with one another and with the layers of context outlined alongside. The potential outcomes of the key mechanisms in certain contexts are shown in the outcomes box on the right-hand side of the model.

The reader will be referred to the explanatory model and individual CMO models as the CMOs are described in the results below.

## CMOs

In the results below, each supporting evidence-informed CMO is summarised under the five overarching areas, including description of key mechanisms, associated and/or potential outcomes, and contexts which interact with mechanisms to produce different outcomes. Consideration of the three RR objectives is integrated into the results. At the start of each overarching area, tables show the titles of supporting CMOs, along with data sources that have informed theory development. Quotations from literature have been included in CMO summaries to support transparency of theory development. A full list of CMOs categorised into context, mechanism (resource and response), and outcomes with multiple data extracts is openly available from Ulster University's Research Portal at DOI [94].

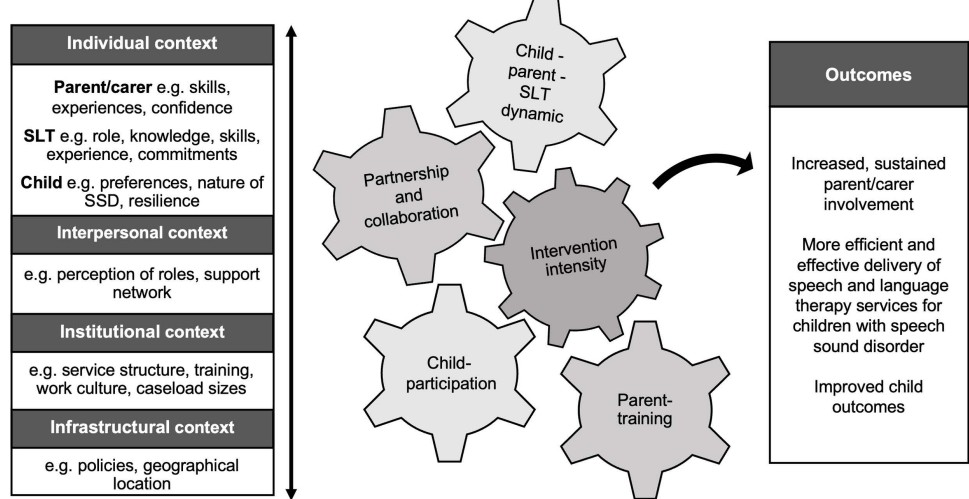

**Fig 2. Explanatory model showing contexts, mechanisms, and outcomes of parent-implemented digital interventions for children with SSD.** This model shows examples of the layers of context involved in digital parent-implemented interventions in the box on the left side. The cogs in the middle of the figure link to the five key programme areas of the intervention: 1. Child-participation, 2. The child-parent-SLT dynamic; 3. Parent-training; 4. Partnership and collaboration; 5. Intervention intensity. The box on the right side of the model shows potential outcomes of the intervention.

**Programme area 1: Child-participation.** Children with severe SSD need to engage with sustained, intensive practise to learn new speech patterns, therefore facilitating child-engagement is paramount in effective parent-implemented intervention. The CMOs titles and evidence sources for this programme area are shown in Table 4.

**CMO 1.1: Adapting the context of home-practice increases child participation.** The timing and environment of home-practice needs to be adapted for each child to help participation, including accounting for competing activities, individual preferences, and the child's temperament. Timing home-practice to suit the child's preferences and needs reduces feelings of exclusion and isolation from favoured activities, possibly reducing resentment towards home-practice. Discussing these contexts with the child gives them a sense of ownership over therapy. Through these mechanisms, the child is more willing to take part, leading to less potential for disruptive behaviours, increased satisfaction for parent/carers and children, and ultimately more frequent home-practice. The mechanism of planning may be dependent on the parent/carers capability to recognise and act on their child's preferences. In a qualitative study of parents' experiences of home-practice (Sugden et al. [29]), they reported:

*Parents also spoke about how their child's behavior and response to home practice influenced the logistics of completing home practice.*

*If I do it with Darcy I'm like "Am I going to have a battle? Am I then going to have a half hour tantrum? Am I ready for that?... And that's where I'd go... actually, no, I'm not, because we're about to go to swimming, we're about to do that, I'm going to have to leave it til later, because if he did have a meltdown, then I'm carrying a child out screaming to the car. (Jane)* (p. 173)

The mobility of devices (i.e., phone or tablet) allows families to practise in different locations, creating more opportunities for choice and capacity to adapt to the child's preferences, which is motivating and possibly more enjoyable for children. Therefore, activities accessed through mobile devices may lead to more sustained practise, increasing the intensity of intervention. It is important to recognise that this mechanism is dependent on access to a mobile device, without which this aspect of the programme would not be possible. The child's motor skills also need to be accounted for to be able to use the functions of mobile devices. Gačnik et al. [72] noted:

*Tablet-assisted therapy was found to enable greater flexibility. The mobility of the device gave the children more options for choosing a favourable body position and was believed to enable practice in more environments. SLPs reported that the children used the tablets in different positions (e.g., placing the tablet on the table or the floor; holding the tablet and/or having the SLP hold it), often changing positions between exercises… (p. 826)*

**CMO 1.2: Individualising task difficulty optimises engagement and learning effect.** This CMO is depicted in Fig 3. Variability within and between children is seen across different activities when considering a child's speech

**Table 4. Programme Area 1: Child-participation.**

| CMO Number | CMO Title | Evidence sources that informed CMO synthesis |
| --- | --- | --- |
| 1.1 | Adapting the context of home-practice increases child participation | [28,44,45,66,70–72,74,75,77,81,84,86–88] |
| 1.2 | Individualising task difficulty optimises engagement and learning effect | [14,66,71,73,77,79,81,84] |
| 1.3 | Offering a sense of social reward increases the child's motivation | [43,45,66,87] |
| 1.4 | Character customisation in digital therapy games motivates more frequent speech practise | [44,45,66,69,72,74–86] |
| 1.5 | Meaningful asset or reward collection in games increases child engagement | [43–45,73,74,81,82,84,87,88] |

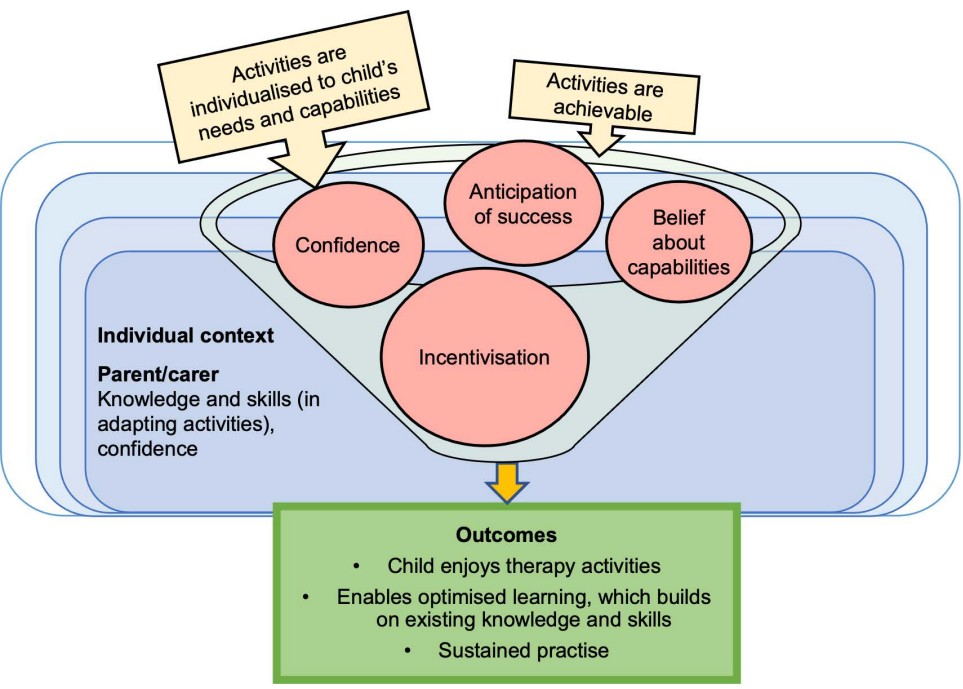

**Fig 3. Model of CMO 1.2.**

sound ability, the impact of their SSD, and their resilience (see child individual context in Fig 2). Literature highlights the importance of the intervening adult tailoring therapy activities, so targets are achievable and appropriate to individual needs and capabilities. In doing this, the child will feel confident to attempt targets because they anticipate success and accomplishment, which incentivises sustained practise and also increases parents/carers satisfaction with home-practice. These attempts will help build on their existing knowledge and skills, leading to optimised child learning, and importantly, enjoyment of activities. The parent/carers' confidence and skills to adapt activities is linked to their self-efficacy. In relation to games, Crowe et al. [66] noted, "*Games that were easy and could be completed more quickly were favourites and facilitated participation while harder games were a barrier*" (p. 278).

**CMO 1.3: Offering a sense of social reward increases the child's motivation.** Studies indicate that digital games well-known to the child and people around them are motivating, as they offer an opportunity to share a common experience with peers and normalise their therapy involvement. Children feel proud of playing the game with a sense of social reward and connection to other people, and they may feel competitive, which is incentivising. These mechanisms may mean children feel more positive towards therapy activities, more motivated to take part, and want to practise more frequently.
Hair et al. [45] noted

*In this way, children could talk about or share their experiences playing the game with their peers, without standing out as different. Children were enthusiastic about playing the game and some seemed very proud of their in-game accomplishments, which we hope they felt free to share with their friends. It could be interesting to explore how reframing speech therapy exercises as a "regular" game changes how they are perceived both by children undergoing therapy and their peers with less exposure to speech therapy.* (p. 21)

**CMO 1.4: Customisation of characters in digital games increases child motivation.** The opportunity to customise characters motivates children to sustain engagement with digital therapy games. Children are enthused and motivated by character customisation, which appears to work by facilitating an emotional connection to games, making it feel meaningful and relatable, as shown in Fig 4. This motivation has the potential to increase the frequency and sustainability of engagement; key to effective speech intervention. Age, gender, previous experience, digital skills, and interest in technology are factors thought to impact on whether children respond with enthusiasm towards digital customisation in games. Game customisation will be especially motivating if it links to children's personal interests.

Silva et al. [88] recognised the impact of digital experience on engagement, noting:

*…three GG [gamification group] subjects (S2, S7 and S9) had knowledge and interest in operating the computer and electronic games. This fact favored engagement throughout the use of the tool in the sessions. In contrast, subjects S3 and S8 (also belonging to the GG) never used a computer and, initially, had difficulties handling the electronic device, making it challenging to engage and motivate with the proposed game immediately.* (p. 6–7)

**CMO 1.5: Meaningful asset or reward collection in games increases child engagement.** Motivating activities support frequent, sustained, and intensive practise. Before or after a speech task, the opportunity to collect chosen assets/rewards to progress through a meaningful game (digital or not) increases child motivation. If rewards are sensitive to individual responses (e.g., receiving rewards for all attempts, but losing rewards when not attempted), children are motivated to sustain attempts. These rewards are thought to give children a feeling of purpose and appreciation of consequence which can be incentivising. Irrespective of the child's awareness of the link between speech tasks and potential asset collection, the mechanism of customised asset collection results in children of all ages wanting to take part in speech activities more frequently, for longer, with more word repetitions, leading to higher intervention intensity. If rewards are given too leniently, this might reduce future motivation.

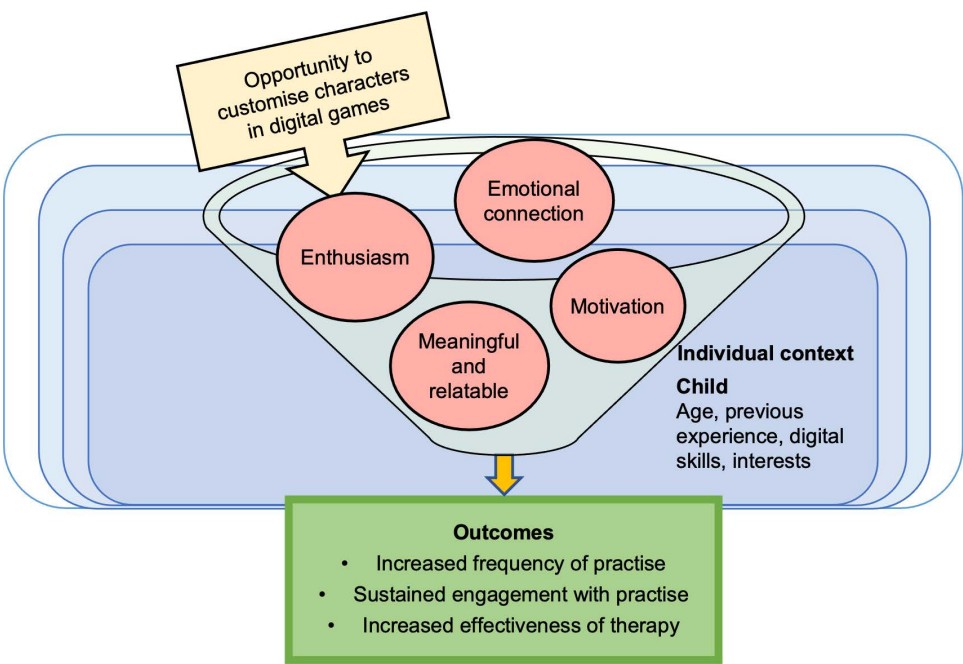

**Fig 4. Model of CMO 1.4.**

McLeod et al. [81] describe the incentivisation gained from rewards when children use their mobile speech app:

*SayBananas! is programmed to incentivise accurate performance in speech exercises. A child is rewarded with points and extra time for completing a speech exercise, but the reward is increased when the response is scored as correct.* (p. 399)

**Programme Area 2: The Child, Parent/Carer and SLT Dynamic.** The dynamic between parent/carer, child, and the SLT may change and evolve over time. This factor moves between acting as a mechanism, context, and outcome, as shown in Fig 2 where this is noted in each area. The CMO titles for this programme area are shown in Table 5.

**CMO 2.1: Home-practice may build understanding between parent/carer and their child.** At the start of therapy, parents/carers may lack confidence and may not allocate time to supporting home-practice. When SLTs ask parents/carers to practise at home, parent/carers are prompted to allocate time to speech tasks with their child. Parent-training may increase parents/carers knowledge of their child's needs, which might increase their level of comfort and confidence to support their child. These new skills and increased exposure to home-intervention can lead to a change in social interaction and connection between parent/carer and child, resulting in more enjoyable practise. It is posited that enjoyable, positive home-practice could increase its frequency. Davies et al. [67] reported on changes in parent's perception of roles, noting:

*For example, one parent explained that she learnt techniques, but also 'completely changed' her interaction with her child, expressed as a transformation in her thinking and behaviour:*

*I was more like a master … I was constantly battling with him, shouting at him, trying to confine him, but now I'm more like a parent … I'm more of his loving parent. (P1, third interview)* (p. 181)

In a study by Sugden et al. [29], one parent reported: *"in some ways it forced me to spend really good time with him, and kind of regularly, and that was really nice and I think really helpful as just kind of a bonding sort of thing"* (p. 171)

**CMO 2.2: Parents/carers may feel uncomfortable adopting an intervenor role.** In parent-implemented interventions, parents/carers are asked to take on the role of intervener, which may not match their skills. Children may not adapt to this change in role, may feel uncomfortable, leading to frustration and reduced willingness to take part (emotions they may be more comfortable to show in a home-setting). The nature of the parent/carer-child relationship may mean parents/carers feel uncomfortable giving negative feedback in tasks, or they may feel conflict about being in a role of parent/carer and therapist at the same time. The response of parent/carer and children to the change in roles may impact on enjoyment of activities and success of child engagement. This may lead to dissatisfaction with delivering parent-implemented interventions at home.

Thomas et al. [37] noted

**Table 5. Programme Area 2: The child, parent/carer and SLT dynamic.**

| CMO Number | CMO Title | Evidence sources that informed CMO synthesis |
|---|---|---|
| 2.1 | Home-practice may build understanding between parent/carer and their child | [29,31,37,67,77,87] |
| 2.2 | Parents/carers may feel uncomfortable adopting an intervenor role | [29,37,67,77,81,87,90] |
| 2.3 | A positive SLT-child rapport may motivate children to practise at home | [29,73,90] |

*Several parents mentioned that their child had less emotional regulation at home with them than in the clinic with the therapist. At home the child expressed more frustration, anger, and annoyance than in the clinic.* (p. 395)

In a study by Sell et al. [87] which explored parents/carers experiences of implementing intervention following training, they reported:

*… there was the challenge of being both teacher and parent, which led to a conflict about their identity. For some parents, there was a struggle with the juxtaposition of parenting while also trying to add the structure of leading articulation therapy and not wanting to distress their child or be seen as constantly disciplining.* (p.974).

They also noted that children may respond differently to their parent's role:

*Parent experience of how their child responded to them varied, with some children finding it difficult to reconcile the parent–child relationship with a teaching relationship, while other children embraced it and loved having the extra engagement with their parent.* (p.976)

**CMO 2.3: A positive SLT-child rapport may motivate children to practise at home.** Data indicates that positive rapport between the SLT and child can be achieved through the SLT showing warmth, honesty, responsiveness, and creating a fun environment (see Fig 5) Understanding the child's 'challenge point' is involved in rapport building, where the SLT facilitates a task that is optimally difficulty; not so difficult that they can no longer process information about the task and not so easy that learning no longer takes place [95]. Through these mechanisms, the child feels a sense of alliance and trust in the therapist (see pink circles in Fig 5). They feel positive about sessions and are motivated to practise at home for the SLT, which is encouraging and incentivising for parents/carers who are keen for their child to have a positive intervention experience. The outcome of positive rapport is improved relationships between child, SLT, and parent/carer, potentially leading to more sustained engagement in parent-implemented therapy (see outcomes in Fig 5).

Watts Pappas et al. [90] reported that *"the parents' wish to ensure that the speech intervention was a positive experience for their child meant that the SLTs' ability to establish professional/child rapport was of paramount importance."* (p. 233)

**Programme area 3: Parent-training.** Literature suggests a crucial link between parent-training and outcomes of parent-implemented interventions in certain contexts. Table 6 shows the CMO titles and evidence sources for programme area 3.

**CMO 3.1: The SLT educates parents/carers about what to do in home-practice, and why.** A key part of parent-implemented intervention is providing education for parents/carers, such as the nature of their child's SSD, intensity of intervention, intervention techniques, and what to do at home. Parent-training may involve opportunities to observe modelling of intervention techniques, allowing parents/carers to see and hear the activity in practice and notice how the child responds. Providing instruction and demonstration increases parents/carers knowledge of what to do, confidence in their skills, understanding the purpose of home-practice, and potentially helps them to implement activities with increased success and fidelity. However, the nature of parent-training should be adapted to meet the parent/carers individual learning context to be effective in achieving these outcomes.

If SLTs communicate with parents/carers about why home-practice is important, parents/carers may realise the purpose and consequence of home-practice, possibly increasing their motivation to engage with the learning. Parent/carers' understanding of the consequences of home-practice may change as intervention progresses and their experience grows. Therefore, those with more experience of implementing intervention at home may be better able to understand the consequences explained to them by the SLT.

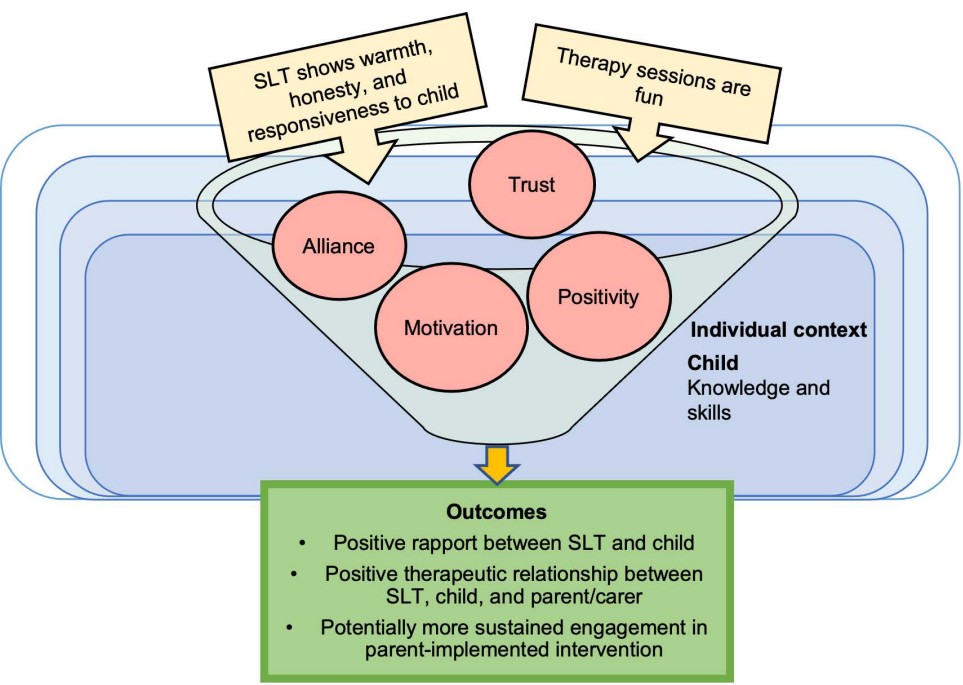

**Fig 5. Model of CMO 2.3.**

**Table 6. Programme Area 3: Parent-training.**

| CMO Number | CMO Title | Evidence sources that informed CMO synthesis |
|---|---|---|
| 3.1 | The SLT educates parents/carers about what to do in home-practice and why | [25,36,71,78,87,89,90] |
| 3.2 | Accessible, individualised explanations improve the effectiveness of home-practice | [23,26,27,29,31,36,67,68,71,73,76,78–81,87,89,90] |
| 3.3 | Parent-training on feedback techniques will support optimal change to the child's speech through home-practice | [23,27,66,69,73] |

In the study by Sell et al. [87] which explored parents' experiences undertaking therapy with their child following training, a parent reported:

*"… it's easier to work with something you understand so I knew why I was doing it...and for the first time ever I knew why he was doing what he was doing and how you could work towards correcting it"* (p.974)

**CMO 3.2: Accessible, individualised explanations improve effectiveness of home-practice.** Data suggests that providing accessible and individualised explanations within the scope of parents/carers capabilities, experiences, and learning styles enables learning. Accessible communication includes using communication methods appropriate for different learning styles to helps parents/carers to see and hear information in a dynamic way, such as, text, pictures, and videos. Tailoring information enables parents/carers learning, increasing confidence with what to do and possibly reducing overwhelm (as shown in Fig 6). Activities are more achievable, which empowers and increases parents/carers' belief in their capabilities. It is also likely that increased success of home-intervention will increase parent/carer and child

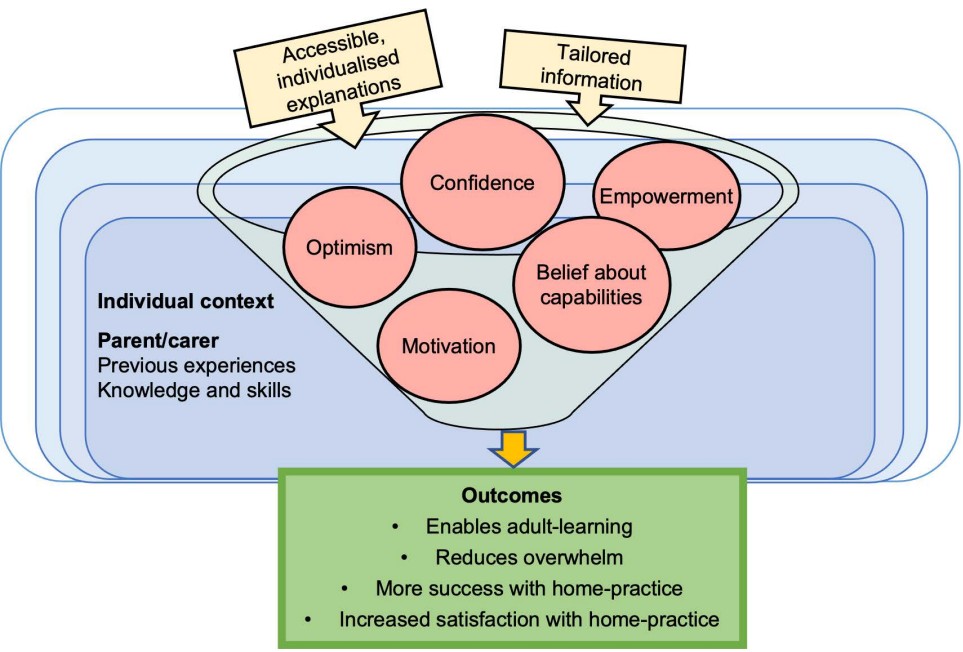

**Fig 6. Model for CMO 3.2.**

satisfaction, which may promote sustained home-practice. These factors are thought to improve the effectiveness of home-practice and outcomes for children with SSD. Hodge and Gaines [73] suggested:

> *…S-LPs need to adapt expectations for parent learning and coaching based upon the parent's incoming skill set (e.g., one parent may be ready for coaching on more advanced skills while a parent who is not understanding concepts of imitation and levels of support using integral stimulation should have additional coaching before moving to more challenging skills).* (p. 52)

**CMO 3.3: Parent-training on feedback techniques will support optimal change to the child's speech through home-practice.** Feedback techniques in speech sound intervention trigger change in a child's knowledge and production of speech sounds, when adapted to suit their existing knowledge and skills. For example, for children with less previous knowledge, tailored feedback gives them insight into their sound production, and they may begin to alter their phonological representations and/or motor programmes. For children with higher levels of existing knowledge, tailored feedback may encourage use of existing phonological knowledge. In either case, the child strengthens their phonological representation and/or motor programme of the sound and produces the sound with greater accuracy and independence on their next attempt.

Parents/carers may lack confidence, may not have the knowledge to give appropriate feedback, and may have different parenting styles. When parents/carers are upskilled through effective parent-training about feedback techniques, they may learn to tailor feedback at home. This gives children more opportunities to receive meaningful feedback, supporting progress with their speech targets. Children may begin to self-correct and generalise their learning. McKechnie et al. [27] noted that *"Variation in response to different feedback types and frequency may be influenced by strength of internal representation of the specific speech behaviors targeted and/or pre- treatment level of proficiency."* (p. 15).

**Programme area 4: Partnership and collaboration.** SLTs and parents/carers identify that a sense of partnership and collaboration has potential to influence responses to parent-implemented interventions for children with SSD. Parent/carer involvement in intervention is complex and multi-faceted. Families have diverse, individual home-lives with different competing demands and priorities. Based on past experiences, parents/carers have varied knowledge or skills, expectations about roles, and beliefs about their capabilities. Similarly, SLTs' expectations of roles, clinical experience, and working contexts vary. Therefore, SLT approaches and parent/carer responses will depend on different layers of context, as portrayed in the model in Fig 2. The CMO titles in this programme area are listed in Table 7.

**CMO 4.1: Empathy and validation from the SLT to the parent may lead to a trusting relationship.** Empathy and validation from the SLT may help build a trusting therapeutic parent/carer and SLT relationship, which may increase engagement in home-intervention (see Fig 7). When SLTs have the awareness, skills, and time to actively ask about, listen to, acknowledge, and embed parent/carer's concerns and feedback, then parent/carers feel respected, supported, and valued by the therapist (see Fig 7, pink circles). As a result, parent/carers have an opportunity to be open, which can increase social connection, trust, and rapport. These factors are thought to increase the parent/carer's sense of self-worth and belief in their capabilities; linked to empowerment. Parental empowerment could increase motivation and willingness to engage in home-intervention, which may increase the amount of practise the child receives (see outcomes in Fig 7).

This has been demonstrated in qualitative interviews with parents/carers:

*Parents also commented on the valuable nature of their own rapport with their SLP [Speech and Language Pathologist], and how this influenced their own response to intervention. Parents valued SLPs who put effort into building this working alliance. For example, Jane felt "it was really important that they were listening to me" about her goals for her child and what she "can and can't do" for homework.* ([29], p. 172)

**CMO 4.2: Tailoring activities to fit with family-life can enable home-practice.** Parent/carers have varied home-lives, priorities, and responsibilities. It can be enabling for parents/carers when SLT's are flexible and responsive to their personal circumstances, by tailoring therapy techniques to suit their needs. Planning with parent/carers to adapt activities to individual routines reduces burden on parents/carers, helping them feel more optimistic about their ability to complete home-practice in the context of existing commitments; supporting engagement. Tailoring activities to fit with the family's life enables therapy activities to be incorporated into daily routines, which may increase the frequency of home-practice to support generalisation and maintenance of progress (see outcomes in Fig 8).

This was captured in the views of parent/carers and SLTs in the literature, e.g.,

*…finding ways to incorporate speech practice into families' regular routine was also a successful strategy for ensuring homework completion (e.g., "Create simple activities that can be completed while completing other daily tasks, like in the car, bath, etc."; "I have found when you are able to include homework with tasks they are already completing as a daily routine, it seems to be less of a burden.").* ([89] p. 1993)

**Table 7. Programme Area 4: Partnership and Collaboration.**

| CMO Number | CMO Title | Evidence sources that informed CMO synthesis |
|---|---|---|
| 4.1 | Empathy and validation from the SLT to the parent may lead to a trusting relationship | [24,29,30,36,71,73,82,85–87,90] |
| 4.2 | Tailoring activities to fit with family life can enable home-practice | [28–30,70,71,86,87,89] |
| 4.3 | Individual and service level contexts impact on SLT confidence and views towards parental involvement | [68,76,86,88,89] |
| 4.4 | Parental doubts about their capability leads to reduced self-efficacy | [14,29,37,67,71,77,86,87] |
| 4.5 | The alignment between SLT and parent about expectations will impact on engagement with home-practice | [14,26,28–31,67,68,71,76,86,90] |

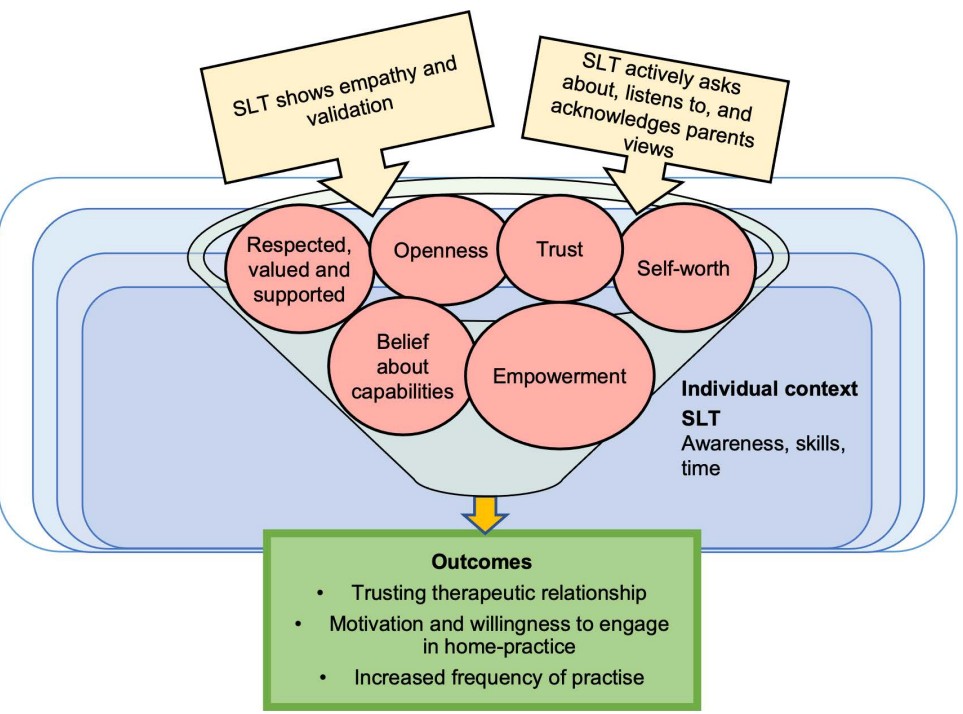

**Fig 7. Model of CMO 4.1.**

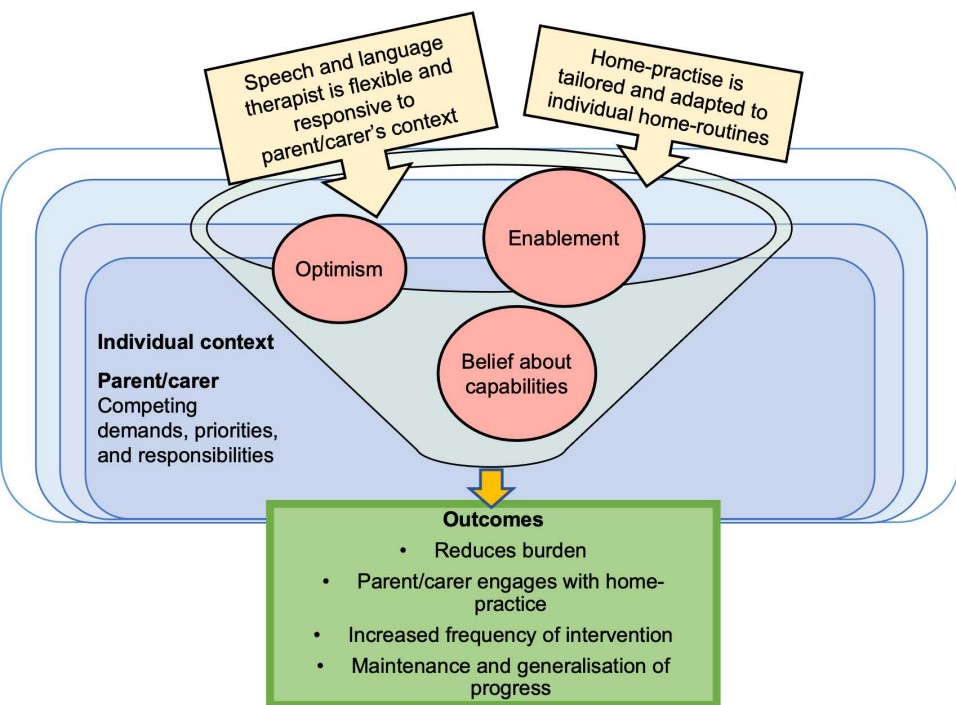

**Fig 8. Model of CMO 4.2.**

**CMO 4.3: Individual and service level contexts impact on SLT confidence and views towards parental involvement.**  Literature from studies in the UK and USA [68,76,89] suggested that SLTs' working contexts, or level of clinical experience, may impact on their involvement of parent/carers in intervention. Certain work-place policies may not facilitate supporting parent-implemented interventions. Newly qualified therapists may not have the confidence or skills to involve parents/carers in intervention. If SLTs have not received adequate training, opportunities, or experience in co-working with parents/carers, they may not realise the benefits, or feel comfortable and confident to involve them. In these contexts, opportunities for parents/carers to take part in intervention to increase the frequency of intervention through home-practice may be missed. Davies et al. [68] noticed:

*The need to include parents as co-workers in intervention appeared to emerge as SLTs gained experience, as illustrated by a recently qualified SLT reflecting on the way she involved parents in assessment:*

*I think her participation was... mmm... as much as it could have been today. Perhaps I could have thought about a different assessment approach, perhaps got her to join in play with the child to see if he communicates differently with her. It's not something I maybe considered before. (SLT 8)* (p. 601)

**CMO 4.4: Parental doubts about their capability can lead to reduced self-efficacy.**  For some parents, being asked to be involved in home-intervention can lead to reduced engagement. If parents/carers have doubts or low confidence about their capability or reduced self-efficacy, being asked to complete home-practice may cause feelings of anxiety or overwhelm, potentially leading them to view home-practice as outside their social role, responsibility, or capability. As a result, parents may adopt a role as an attender (rather than intervener), leading to reduced opportunity for home-practice outside SLT-led sessions.

Sell et al. [87] explored parents experiences of delivering intervention for their child with cleft palate speech disorders following training. One parent stated:

*" …we were a bit nervous about really doing much with him at home because we didn't want to confuse him or get it wrong because we just didn't feel we understood what the therapist was doing"* (p.974)

**CMO 4.5: The alignment between SLT and parent/carer about expectations will impact engagement with home-practice.**  Setting expectations about intervention roles can support effective partnership working and parent-involvement, particularly if established at the beginning of the therapy process when there may be limited shared understanding of roles. If the SLT asks about and listens to parental expectations and can facilitate open and honest conversations and negotiation about how roles will evolve, a mutual understanding of roles can be developed, and possible misconceptions can be resolved early on. This will result in effective co-working and partnership where parents/carers may be more likely to adopt an intervener role, potentially increasing the frequency of home-practice and leading to better therapy outcomes. Davies et al. [68] explored SLTs' conceptions of roles, and one SLT noted:

*"In the past we've not helped ourselves by this air of mystique or that these children are going to come and then we're going to fix them. Yes, we've always given them homework, but as I say, we're much better at setting out our stall outright at the beginning and saying this is what we do, how we work." (SLT 3)* (p. 602)

Due to previous experiences, parents/carers and SLT's role expectations may not align. A continued lack of shared understanding of roles, particularly where there is conflict between parent/carers expectations and being asked by the SLT to be involved in intervention delivery, could mean parents/carers feel unprepared or unwilling to take part. Parents/carers may distrust the SLT if they perceive them to have unfulfilled their responsibility as the sole intervener. These responses

may cause disengagement from parent-implemented intervention and dissatisfaction with the intervention offered, e.g., in Watts Papas et al. [90] one parent noted, "*well, I don't see how it can be a partnership when you're dealing with professional people. It's their job, you can't tell them what to do. You can't take your car to the mechanic and be a partner with the mechanic, can you?*" (p. 236)

**Programme area 5: Intervention intensity.** Intervention intensity is paramount to effective speech sound intervention. In certain contexts, mechanisms in this area are associated with greater, faster, and more sustained child progress. In other contexts, outcomes may be increased sense of burden, worry and dissatisfaction with support received. Table 8 outlines the CMOs and evidence sources identified in this programme area.

**CMO 5.1: More frequent intervention sessions lead to better therapy outcomes.** Data suggests that more frequent sessions help children build on learning and existing knowledge from previous recent practise to promote deep, enhanced learning, therefore reducing need for re-teaching in subsequent sessions. The outcome is greater and faster, impacting sustained progress, functional gains, and reduction of the overall duration of therapy. These intervention effects may lead to more effective and efficient service provision. This CMO is depicted in Fig 9. This is drawn from data from several papers, for example, Cummings et al. [9] noted, "*more frequent intervention sessions could reduce the need for re-teaching because children are more likely to remember the skills practiced in a session that recently occurred.*" (p.112)

The mechanism of frequent appointments sits within a backdrop of contextual challenges for SLTs in providing opportunities for regular clinic sessions due to service-level constraints, and for parents/carers to commit to regular appointments. Findings demonstrate that home-intervention can supplement direct SLT input to give children opportunities to practise regularly, for example at least more than once a week with shorter gaps. Frequent practise reinforces skills across different every-day environments, helping habitualisation, generalisation, and maintenance of new skills across different settings. As a result of home-practice, children receive a higher intensity of intervention, leading to an increased intervention effect, and potentially reducing costs to services. Tosh, Arnott & Scarinci [31] suggest:

*…it is reasonable to anticipate that if parents can be effectively trained to deliver HP [home programme] interventions they can then deliver a higher dosage of ongoing therapy at minimal ongoing cost to the service provider thereby providing a better long-term cost-effectiveness than TT [traditional therapy].* (p. 266)

**CMO 5.2: Life-factors impact on parent/carer response to highly frequent sessions.** Twice-weekly sessions (as often targeted in the research) can be difficult for some families to engage with. Families have other extraneous commitments and barriers. The greater the competing commitments, the more difficult it is to engage in home-practice or attend regular sessions. When the request for parents/carers to attend twice-weekly clinic sessions and/or to practise at home does not fit with other commitments or barriers, parent/carers may feel a sense of burden, worry, and overwhelm, possibly resulting in dissatisfaction with services and de-motivation to take part. Consequently, the family does not, or is unable to engage in frequent sessions, and the child misses an opportunity to receive intensive intervention (see Fig 10).

**Table 8. Programme Area 5: Intervention Intensity.**

| CMO Number | CMO Title | Evidence sources that informed CMO synthesis |
|---|---|---|
| 5.1 | More frequent intervention practise leads to better therapy outcomes | [4,9,11,14,24–29,31,37,71,78–83,85] |
| 5.2 | Life-factors impact on parent/carer response to highly frequent sessions | [9,11,25,26,28,29,37,67,70–72,77,86] |
| 5.3 | Higher dose influences clinically significant change | [8,9,11,81–83,85,87] |
| 5.4 | Providing therapy of optimal continuous intervention duration produces better outcomes | [4,9,73,81,85] |

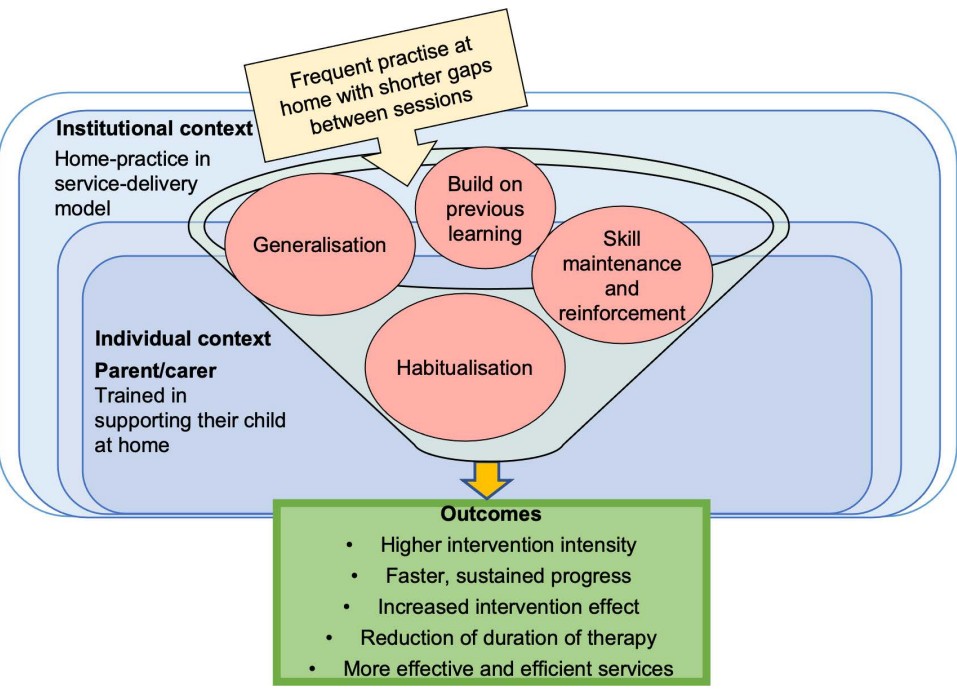

**Fig 9. Model of CMO 5.1.**

This parental experience is described by Thomas et al. [37]

*In some cases the challenges associated with accessing therapy became overwhelming. For Tracy [parent], the barrier of distance and associated travel time resulted in Lachlan with-drawing from his community therapy:*

*It was an hour one-way to get to therapy. And to only be there for sort of half an hour or 45 min, it was too much. So we put a halt to it there. (Tracy)* (p. 394)

**CMO 5.3: Higher dose influences clinically significant change.** Optimal dose (speech production practise) is a key mechanism in effective speech sound intervention. When children are provided with a dose in sessions through rehearsal and repetition of targets, suited to the nature of their SSD and approach selected, the child will engage in deep learning resulting in a learning effect. This potentially results in clinically significant change and improved child outcomes, thought to lead to more efficient and effective Speech and Language Therapy services.

A recent Quality Improvement project by McFaul et al. [11] noted, *"Consideration of the possible change mechanisms underpinning the success of high intensity intervention for children with severe SSD suggests that higher doses within sessions reach a threshold which supports an optimal practice/learning effect."* (p. 1).

Namasivayam et al. [82] (2023) noted how dosage impacts on outcomes for children with motor-based SSD:

*Too high of a dose results in overlearning (continuing practice after skill mastery), whereas too low of a dose results in underlearning (practice termination before criterion performance has occurred).* (p.11)

**CMO 5.4: Providing therapy of optimal continuous intervention duration produces better outcomes.** The overall length of therapy, tailored to the child's individual needs, is a key active ingredient. Receiving therapy over an optimal

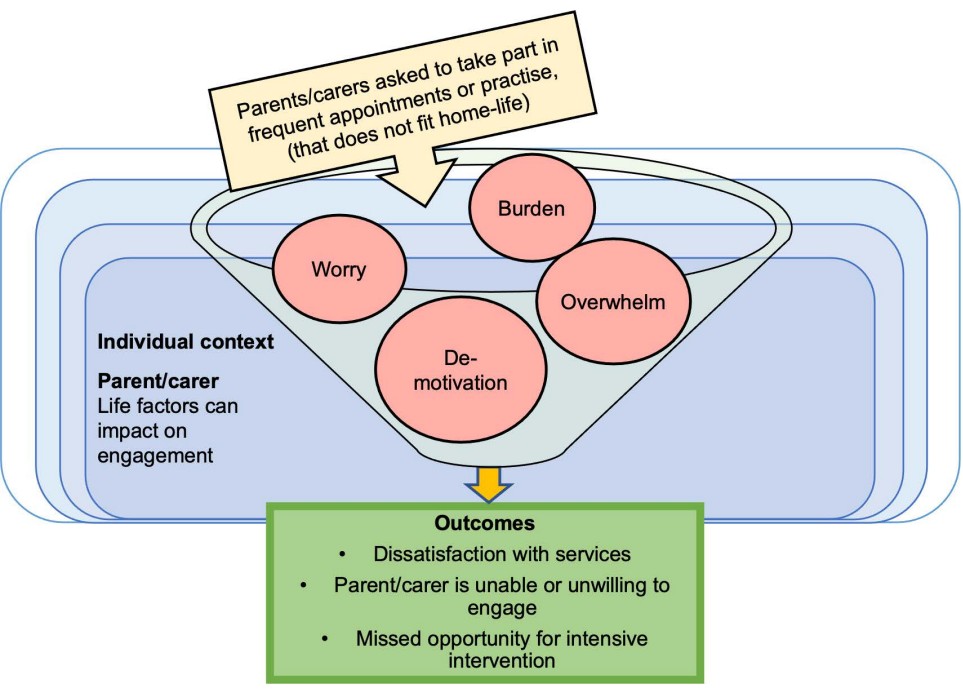

**Fig 10. Model of CMO 5.2.**

continuous duration means intervention is long enough (if delivered with the optimal frequency) to help the child to learn and maintain new skills, but not longer than is necessary. This helps the child to consolidate and retain their learning in sessions, leading to greater intervention effects. Allen [4] noted, *"These different findings may indicate that a duration of six sessions is enough to significantly impact phonological skills, but a greater total intervention duration—for example, 24 sessions—is needed to demonstrate an intensity effect [generalised learning]"* (p. 873).

## Discussion

This realist review aimed to build on understanding of how parent-implemented intensive interventions work, for who, why, and in what situations. We developed 20 theoretical evidence-informed explanations using inferences and theoretical causal claims supported by extracted data from the papers reviewed which were underpinned by the BCW [91] and TDF [60,61], self-efficacy theory [92], and adult-learning theory [93]. This thinking informed the development of the final overarching explanatory theoretical model as proposed in realist reviews (see Fig 2). Key behavioural change mechanisms (which in realist science include the behavioural change resources provided and the response to those resources) were identified to enhance effectiveness of digital intensive parent-implemented interventions. These mechanisms weave through all five programme areas as depicted in the explanatory model and include: customisation to suit service-users; developing trusting relationships and partnerships; and supporting motivating opportunities for intensive intervention. This review offers insight into how context may impact on these key active ingredients to produce different outcomes (see Fig 2).

Our findings sit within existing research on intensive, parent-implemented, digital interventions in speech and language therapy. A core aspect of thinking developed through this review is around how and why increased, optimally intensive intervention, supports greater, more sustained progress for children with SSD. Underpinning mechanisms of deep, enhanced learning from repetition and rehearsal help explain how intensity may work. The CMOs highlight how digital, parent-implemented interventions for children with SSD have the potential to support parents/carers to deliver intervention

at home, creating opportunities for more frequent practise with shorter gaps between sessions. This could offer significant contribution to services, where highly frequent sessions are challenging to deliver in current contexts.

The Theoretical Domains Framework (TDF) [60,61] supported explanation of mechanisms and contexts. Key domains identified included: cognitive and interpersonal skills; knowledge; belief about capabilities; belief about consequences; social or professional role or identity; and environmental context and resources. These can be seen in Fig 2 (as mechanisms and contexts) and will be referred to in this discussion.

In line with existing reviews [31,36], belief about capabilities (a domain of the TDF), is a response linked to motivation and empowerment, generally promoting active engagement of parents/carers, children, and SLTs in intervention. In self-efficacy theory [92], empowerment relates to an individual's perception of competence and ability to achieve success. This review indicates that empowerment and belief about capabilities is enhanced by parents/carers feeling they are given time, acknowledgement, and respect from the SLT, and by activities being tailored to meet individual contexts (i.e., knowledge, skills, competing demands).

Children's, parent/carers', and SLT's expectations, experiences, and extraneous commitments impact on their belief about their capability and self-efficacy. People are more likely to engage with activities when they believe they will be successful. When asked to complete home-practice, those with a greater sense of self-efficacy may respond with optimism, whereas reduced self-efficacy may lead to feelings of worry or overwhelm. Lower self-efficacy may reduce the response to learning or relationship-building opportunities. As in Fig 2, outcomes (e.g., service satisfaction) will change dependent on this context.

The effects of social role and identity (as in the TDF) is important in understanding contexts of parent-implemented interventions, and how they may impact on child/family/SLT engagement. Reduced shared understanding of roles can lead to reduced satisfaction, trust, and involvement with parent-implemented interventions. Setting and negotiating clear expectations with parents/carers can help parental empowerment and engagement. Children and parental roles may change through parent-implemented interventions. Family responses to this will depend on past experiences, expectations, and skills (shown as individual contexts in Fig 2).

Our findings corroborate those from existing literature on parent-implemented interventions, where parent-training is a key factor. SLT and parent/carer accounts about supporting parental learning are underpinned by Knowles' principles of adult-learning theory [92]. Parents/carers are most likely to learn when they understand why information is important and has personal relevance to their life and goals. Training that is adapted to suit existing experiences provides a platform for future learning. SLTs need to be aware of adult-learning principles and have skills to adapt communication to suit individual needs, enabling empowerment of parent/carers, as seen in SLT individual contexts in Fig 2.

Across all aspects of the intervention, the individual context of children and their families requires careful consideration. As in Brofenbrenner's ecological systems theory [96] the child's needs sit within the context of their family and community, and an impact on one will impact on another. The associated demands of delivering digital, intensive, and parent-implemented intervention need to be matched to children's and families' competing commitments, capabilities, knowledge, experiences, personal motivations, and goals (see individual contexts in Fig 2). When individual contexts are accounted for, beneficial outcomes of specific mechanisms can be seen, including engagement with home-intervention, satisfaction with services, active and sustained involvement in therapy, intervention fidelity, and greater intervention intensity; associated with increased intervention effect and more effective use of services [9,11].

This review indicated that digital and traditional speech interventions share similar mechanisms, for example, personalisation and rewards are motivating for children, as are digital games which mimic real life games [43]. Findings suggest a digital tool can offer a platform to address certain contextual challenges in accessing intensive parent-implemented speech interventions, including increasing flexibility, mobility, motivation, and incentivisation. Children's individual contexts can impact on these mechanisms and outcomes and need careful consideration, for example, children's age, knowledge, physical skills, experience with digital games, and access to digital resources, some of which could act as contextual barriers to intended outcomes (see individual context examples in Fig 2). It is important to note that digital tools are just

that, tools, and their successful use to support intensive parent-implemented intervention will be dependent on the interaction of all the key mechanisms of change reviewed above. Importantly, differences in intervention outcomes (whether supported by a digital tool or not) are also very reliant on context as shown in Fig 8 and Fig 10.

Wider service-level contexts form a backdrop to the delivery of intervention for children with SSD. Organisational cultures can present contextual barriers or facilitators to successful implementation of interventions. There are clear challenges providing the recommended intervention intensity for children with SSD within clinical contexts, with discrepancies between services and geographical areas due to caseload sizes, workloads, and service structure (see institutional and infrastructural contexts in Fig 2). Explanatory theories in this review have shown that parent-implemented intervention can help increase intervention intensity; SLTs can help increase parental confidence and empower parents/carers to implement intensive intervention at home with their child through offering training and support alongside empathy, flexibility, understanding and partnership. Understanding the key active ingredients involved in effective parent-implemented interventions for this client group, and how they work in different contexts, is paramount so that we can sustainably develop and implement evidence-based intervention intensity in practice and help improve the efficiency of services.

## Clinical implications

This realist review offers new, shared understanding about the active ingredients involved in digital, intensive, parent-implemented interventions for children with SSD. Our findings show how this approach can facilitate more intensive SSD intervention when combined with direct SLT input. Digital tools have potential to incentivise families and increase access to home-practice. However, the review has highlighted that additional underlying active ingredients facilitated by the SLT and parent/carer are needed for digital tools to work.

SLTs need to be aware of the importance of working collaboratively with parents/carers, as developing trust and partnership creates the foundation of the intervention. In successful collaboration, SLTs need to support development of shared expectations and listen to, acknowledge and validate parents'/carers' needs to help build self-efficacy and empowerment. This needs consideration from SLT services and Higher Education Institutes to ensure SLTs are trained and upskilled to work effectively with parents/carers.

This study highlights how individual family contexts can influence the effectiveness of parent-implemented interventions. SLTs need to consider who the intervention will be accessible and appropriate for, and when, considering individual and personal contexts highlighted in this review such as competing family demands, time, parental confidence, home environments, and child temperament and preferences. Scope is needed for SLTs to be flexible in their approach, so that they can individualise the intervention to account for home-circumstances, experiences, and existing knowledge and skills. Adult-learning theory needs consideration when training parents/carers, including supporting understanding of the purpose of home-practice, tailoring information to individual experiences, and facilitating learning through using different modes of communication, demonstration, and modelling. These implications need careful thought in routine practice to optimise the outcomes of digital, intensive, parent-implemented interventions.

## Limitations

For the scope of this review, we included literature that was published post 2012, written in English, which offered specific insight into intervention for children with SSD. It may be that new or different insight would be added from expanding the timescale or client groups included. While the included papers added insight into child and family circumstances and contexts that may impact on outcomes of digital, parent-implemented interventions, the level of detail reported in studies about participant demographics was mixed, particularly in relation to SES and diversity. Some studies reported inclusion of participants with varied demographic characteristics (e.g., [14,29,77–79]), however, where reported, this data was typically not used in interpretation of results. Certain studies acknowledged that using demographic data to evaluate findings is required in future research (e.g., [14,79]). It is understood that digital, parent-implemented interventions will not work

for all families and children with SSD, and further consideration of how it may work for groups that are not represented in current literature included in this review is required to add depth to this understanding.

It is acknowledged that in realist approaches, the conclusions drawn could be dependent on the researcher's interpretation of available data. This was minimised through involving several researchers in a rigorous selection and analysis process, which was clearly and transparently documented. An expert steering group were involved in the development of search terms and initial rough programme theories, helping capture insight from people with lived experience.

This review has developed theories (CMOs) about how the intervention works based on existing literature available. Future research through a realist evaluation could further test (e.g., confirm, refute, or refine) theories through gathering perspectives of stakeholders (e.g., SLTs and parents/carers) about how this intervention would work in clinical practice. In the literature included in this review, there is limited data that considers the child's voice directly. It is important that future research in this area carefully considers how the child's perspective can be captured.

## Conclusion and future directions

Parent-implemented digital interventions for children with SSD can offer a way of increasing intervention intensity to meet evidence-based recommendations and increase the efficiency and effectiveness of therapy. Different layers of context impact on outcomes of this approach, which must be considered in implementation. Digital tools offer a platform to support intensive, parent-implemented interventions and can reduce contextual barriers, however, direct SLT support and partnership with parents/carers are needed to facilitate the key active ingredients of the intervention. Future research should gather perspectives of stakeholders to further test and refine the developed theories. Findings from this review and a future realist evaluation will inform the co-production of a digital tool to support intensive home-practice for children with SSD.

## Supporting information

**S1 Fig. Flowchart of Realist Review Methods.**
(TIF)

**S2 Fig. Example CMO model.**
(TIF)

**S3 Table. Search terms.**
(PDF)

**S4 Table. List of included studies and key study characteristics.**
(PDF)

**S5 Text. RAMESES Standards Checklist.**
(PDF)

## Acknowledgments

We would like to thank the expert steering group who have been involved in this project. We are grateful for their invaluable support with the planning of the review, insight into theory building, and dissemination of results.

## Author contributions

**Conceptualization:** Naomi Leafe, Emma Pagnamenta, Laurence Taggart, Mark Donnelly, Angela Hassiotis, Jill Titterington.

**Formal analysis:** Naomi Leafe.

**Investigation:** Naomi Leafe.

**Methodology:** Naomi Leafe, Emma Pagnamenta, Laurence Taggart, Mark Donnelly, Angela Hassiotis, Jill Titterington.

**Project administration:** Naomi Leafe.

**Resources:** Naomi Leafe.

**Supervision:** Emma Pagnamenta, Laurence Taggart, Mark Donnelly, Jill Titterington.

**Validation:** Naomi Leafe, Emma Pagnamenta, Jill Titterington.

**Visualization:** Naomi Leafe, Jill Titterington.

**Writing – original draft:** Naomi Leafe.

**Writing – review & editing:** Naomi Leafe, Emma Pagnamenta, Laurence Taggart, Mark Donnelly, Angela Hassiotis, Jill Titterington.

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
