## [Decision Letter · Decision Letter 0]

24 Oct 2024

PONE-D-24-32348What works, how and in which contexts when using digital health to support parents/carers to implement intensive speech and language therapy at home for children with speech sound disorder? A realist reviewPLOS ONE

Dear Dr. Leafe,

Thank you for submitting your manuscript to PLOS ONE. After careful consideration, we feel that it has merit but does not fully meet PLOS ONE’s publication criteria as it currently stands. Therefore, we invite you to submit a revised version of the manuscript that addresses the points raised during the review process.

Please submit your revised manuscript by Dec 08 2024 11:59PM**.** If you will need more time than this to complete your revisions, please reply to this message or contact the journal office at plosone@plos.org . Please include the following items when submitting your revised manuscript:

We look forward to receiving your revised manuscript.

Kind regards,

Professor Sandeep Reddy

Academic Editor

PLOS ONE

Journal Requirements:

"This review has been completed as part of a PhD studentship at Ulster University (NL) which is supported by the Department for the Economy (DfE).

3. In the online submission form, you indicated that [The data underlying the results presented in the study are available from Professor Laurence Taggart (l.taggart@ulster.ac.uk).]. 

Reviewers' comments:

Reviewer's Responses to Questions

**Comments to the Author**

1. Is the manuscript technically sound, and do the data support the conclusions?

Reviewer #1: Yes

Reviewer #2: Yes

2. Has the statistical analysis been performed appropriately and rigorously? 

Reviewer #1: N/A

Reviewer #2: N/A

3. Have the authors made all data underlying the findings in their manuscript fully available?

Reviewer #1: Yes

Reviewer #2: Yes

4. Is the manuscript presented in an intelligible fashion and written in standard English?

Reviewer #1: Yes

Reviewer #2: Yes

5. Review Comments to the Author

Reviewer #1: The method is robust and well-structured, uses the realist review approach to explore the complexities of parent-implemented speech and language interventions. Moreover, the use of a realist review is particularly commendable. It goes beyond traditional review approaches by focusing on understanding the underlying mechanisms of how, why, and in what contexts these interventions are most effective.

Reviewer #2: Thank you for the opportunity to read this article. I found this paper of great relevance to clinical practice and the field of research. I congratulate the authors on a well-organised, clearly written paper with excellent objectives that hold importance and application to modern day work with families. Tables 4 to tables 8 alone are a fantastic resource for the field internationally. It is clear a lot of work has gone into this publication.

The only question I feel you have not answered is for whom (see abstract, line 41) – for which parent/carers and for which children with SSD might these parent-implemented digital innovations work for? Do all your papers sample white, middle-high class, well educated, English speaking mothers? That’s an important lens to raise in your conclusion and in your limitations as it has implications for how SSD interventions will work with underserved / under researched groups, i.e., fathers, global majority families, low SES families living in and out of western, education, industrialised, rich and democratic (WEIRD) countries.

Further, whilst I appreciate you have a published protocol, I feel a bit more detail is required in this manuscript. Not everyone, especially clinicians and non-SLT readers, will have capacity to read both and you are doing your brilliant work a disservice by not including single sentences to describe important decisions such as your exclusion/inclusion criteria.

I feel these suggestions will strengthen the paper:

Introduction

1 Strong introduction that clearly builds your argument for the work. As PLOS ONE is a generalist journal, I suggest a definition of SSD, levels of incidence, and perhaps a descriptor of how a child with a ‘moderate’ or ‘severe’ SSD would present.

2 Table 1, page 5. Please share some participant demographics (child age, SES, languages spoken) on these studies – see point above ‘for whom’.

3 Line 166, p 8 of the manuscript: ‘Bellon-Harn et al. [29] studied the use of videos/digital media in parent167 implemented interventions for children with primary SSD or language disorder’. Can you add HOW Bellon-Harn and colleagues did this, i.e. was it an RCT? etc.

Method

4 Add one sentence on why your team selected a realist review it as a method to this manuscript

5 P 10. Again, for time-poor readers unable to read your protocol paper, I would add your inclusion and exclusion criteria for papers to your methods. I only noticed your criterion in your supplementary files, i.e., ages 2-7, no comorbidities.

6 p. 14, line 243 ‘explicit or implicit references to relevant MRTs were noted, then explored in depth, and tested using evidence extracted from the literature’. Please provide one example of an MRT. Perhaps better added in Table 2 – as you provide examples of other definitions here already.

7 p.15, line 258. I wondered if one or two of your IMRTs could be listed, so early thinking is shared. I don’t see these in your protocol with a quick glance.

8 Your s1 flowchart labels ‘PPI’ but your manuscript does not refer to PPI. Please add a PPI paragraph to describe those involved and how they shaped the review. Did they refine your RQ for example? Who were they? Also, is PPI the same as your expert steering group? Who were they? Any experts with lived experience?

Results

9 I would perhaps consider adding the table of included publications to the paper (rather than a supplementary file) and to save space, remove the title of the paper and instead report on some other characteristics such as country of study, research design, sample size of each study, SES, quality rating, maybe even some cross referencing to tables 4 – 8 i.e. which papers contributed to which CMOs (i.e. paper 1 Amed et al 2018 – CMOs 1.1, 1.4, etc)

10 I would also consider listing your CMOs/programme areas in a different priority – first centring the child (participation), then parent, parent/SLT collab, and ending with intensity. This would be in line with Brofenbrenner’s theory of the child first, within a context of their family and community.

Whilst reading through CMO 2.2 I kept wanting to read about the variable partnership between parent and child with SSD? What about their interaction, communication, temperament when without the SLT present and trying to work on SSD Tx? Much later, I then read CMO 4.2. It made me think intensity is irrelevant if the child will not participate, hence the re-structure suggestion.

11 Fig 2 – Are the cogs different sizes because they hold different weight/significance? Can you number the cogs so they match up with your programme areas?

12 For ‘CMO 2.3: Individual and service level contexts impact on SLT confidence….’ You say ‘Literature suggested that SLTs’ working contexts…’ – can you qualify – in which countries? I sense UK studies, but is this a global (WEIRD countries) phenomenon?

Discussion

13 It would be great to see more of the wider literature on digital health tools / behaviour change. Could it be worth looking at Behaviour Change Techniques and Mechanisms of Action and weaving some of these through your discussion – see https://pmc.ncbi.nlm.nih.gov/articles/PMC6636886/

For example - P. 37, Line 795-796 ‘…how behaviour change theories (which ones, add references) and adult-learning theory (which ones, add references) underpin this complex intervention’.

Limitations

14 See first point – you mention ‘for whom’ but I think more detail needs to be added here, rather than ‘all parents’ and ‘all children with SSD’ – is there bias within the recruitment of your included studies?

6. PLOS authors have the option to publish the peer review history of their article (what does this mean? ). If published, this will include your full peer review and any attached files.

**Do you want your identity to be public for this peer review?** For information about this choice, including consent withdrawal, please see our Privacy Policy .

Reviewer #1: No

Reviewer #2: **Yes: ** Martina Curtin

---

## [Author Response · Author response to Decision Letter 1]

20 Feb 2025

Thank you for taking the time to review our manuscript and for the constructive and helpful comments from the reviewers. Please refer to the response to reviewers letter that has been uploaded with this submission, which includes a response to each comment made by the editor and reviewer comments and outlines the relevant revisions made.

---

## [Decision Letter · Decision Letter 1]

10 Mar 2025

What works, how and in which contexts when using digital health to support parents/carers to implement intensive speech and language therapy at home for children with speech sound disorder? A realist review

PONE-D-24-32348R1

Dear Dr. Leafe,

We’re pleased to inform you that your manuscript has been judged scientifically suitable for publication and will be formally accepted for publication once it meets all outstanding technical requirements.

Kind regards,

Roberto Vagnetti

Academic Editor

PLOS ONE

Additional Editor Comments (optional):

Dear Naomi Leafe,

I hope this email finds you well.

The review of the manuscript has been completed.

The reviewer has indicated that all concerns have been addressed satisfactorily, and they consider the manuscript suitable for publication.

Sincerely,

Dr Roberto Vagnetti

Academic Editor

Reviewers' comments:

Reviewer's Responses to Questions

**Comments to the Author**

1. If the authors have adequately addressed your comments raised in a previous round of review and you feel that this manuscript is now acceptable for publication, you may indicate that here to bypass the “Comments to the Author” section, enter your conflict of interest statement in the “Confidential to Editor” section, and submit your "Accept" recommendation.

Reviewer #1: All comments have been addressed

Reviewer #2: All comments have been addressed

2. Is the manuscript technically sound, and do the data support the conclusions?

Reviewer #1: Yes

Reviewer #2: Yes

3. Has the statistical analysis been performed appropriately and rigorously? 

Reviewer #1: N/A

Reviewer #2: N/A

4. Have the authors made all data underlying the findings in their manuscript fully available?

Reviewer #1: Yes

Reviewer #2: Yes

5. Is the manuscript presented in an intelligible fashion and written in standard English?

Reviewer #1: Yes

Reviewer #2: Yes

6. Review Comments to the Author

Reviewer #1: (No Response)

Reviewer #2: Huge congratulations on a fantastic piece of work, that provides new internationally-relevant learning to the fields of practice and research.

7. PLOS authors have the option to publish the peer review history of their article (what does this mean? ). If published, this will include your full peer review and any attached files.

**Do you want your identity to be public for this peer review?** For information about this choice, including consent withdrawal, please see our Privacy Policy .

Reviewer #1: No

Reviewer #2: **Yes: ** Martina Curtin

---

## [Editor Report · Acceptance letter]

PONE-D-24-32348R1

PLOS ONE

Dear Dr. Leafe,

I'm pleased to inform you that your manuscript has been deemed suitable for publication in PLOS ONE. Congratulations! Your manuscript is now being handed over to our production team.

Kind regards,

on behalf of

Dr. Roberto Vagnetti

Academic Editor

PLOS ONE